# Enose Lab Made with Vacuum Sampling: Quantitative Applications

**Guilherme G. Teixeira** [1] , **António M. Peres** [1] , **Letícia Estevinho** [1] , **Pedro Geraldes** [2] , **Cristina Garcia-Cabezon** [3] , **Fernando Martin-Pedrosa** [3] , **Maria Luz Rodriguez-Mendez** [4] and **Luís G. Dias** [1,*]

[1] Centro de Investigação de Montanha (CIMO), Instituto Politécnico de Bragança, 5300-324 Bragança, Portugal; guilhermegob@gmail.com (G.G.T.); peres@ipb.pt (A.M.P.); leticia@ipb.pt (L.E.)

[2] Instituto Politécnico de Bragança, 5300-324 Bragança, Portugal; pedrogeraldes@ipb.pt

[3] Department of Materials Science, Universidad de Valladolid, 47011 Valladolid, Spain; crigar@eii.uva.es (C.G.-C.); fmp@eii.uva.es (F.M.-P.)

[4] Group UVaSens, Escuela de Ingenierías Industriales, Universidad de Valladolid, Paseo del Cauce, 59, 47011 Valladolid, Spain; mluz@eii.uva.es

\* Correspondence: ldias@ipb.pt; Tel.: +351-96-802-7957

**Abstract:** A lab-made electronic nose (Enose) with vacuum sampling and a sensor array, comprising nine metal oxide semiconductor Figaro gas sensors, was tested for the quantitative analysis of vapor–liquid equilibrium, described by Henry's law, of aqueous solutions of organic compounds: three alcohols (i.e., methanol, ethanol, and propanol) or three chemical compounds with different functional groups (i.e., acetaldehyde, ethanol, and ethyl acetate). These solutions followed a fractional factorial design to guarantee orthogonal concentrations. Acceptable predictive ridge regression models were obtained for training, with RSEs lower than 7.9, $R^2$ values greater than 0.95, slopes varying between 0.84 and 1.00, and intercept values close to the theoretical value of zero. Similar results were obtained for the test data set: RSEs lower than 8.0, $R^2$ values greater than 0.96, slopes varying between 0.72 and 1.10, and some intercepts equal to the theoretical value of zero. In addition, the total mass of the organic compounds of each aqueous solution could be predicted, pointing out that the sensors measured mainly the global contents of the vapor phases. The satisfactory quantitative results allowed to conclude that the Enose could be a useful tool for the analysis of volatiles from aqueous solutions containing organic compounds for which Henry's law is applicable.

**Keywords:** electronic nose; MOS sensor array; quantitative analysis; ridge regression

## 1. Introduction

An electronic nose (Enose) is a device that can detect volatile compounds based on the signals recorded by a sensor array, mimicking the human nose's behavior. It has been applied in several areas such as health care, environmental monitoring, industrial applications, automobile, food storage, and military applications [1].

There are three main steps to be considered in a volatile analytical system: preparation of samples and their sampling procedure; measurement method; data analysis [2]. Sampling is performed by exposing the sensors' sensible materials to the volatiles generated on the headspace of a flask containing a target solution, until the signals reach equilibrium [3] or by delivering the vapor sample of the headspace flask to the sensors array [4]. Cleaning of the sensors surface is an important issue in order to provide a stable baseline, usually achieved by using a reference gas (mainly normal air) that allows to recover a constant signal value for each sensor [5]. Metal oxide semiconductors (MOS) sensors have been the most widely used for building sensor arrays due to the fact of their high sensitivity, low response time, excellent cost–benefit, portability, accuracy, stability and durability [6,7]. MOS comprise a set of electrodes with different sensibilities that together generate a "fingerprint" corresponding to the samples' volatile characteristics. The fingerprint is the

result of the changes in the sensors' properties due to the fact of chemical reactions and/or physical interactions, which take place when the sensors' sensitive element is put in contact with the sample's volatile content. These changes are converted into electrical signals by a transduction system [8], being then treated by selective computerized multivariate statistical data processing tools, which can evidence qualitative or quantitative variations in the composition of the sample [5]. Fingerprints are highly influenced by the sampling methodology [6,9]. Industrial applications of MOS-based Enoses have been reported for the analysis of indoor air quality [10], flammable liquids [11], wastewater and water treatment processes monitoring [12,13], quality of beverages [14], ethanol in beverages [15], discrimination of products [16], and the evaluation of soil moistures [17], but they are mainly used in the food sector [18–20].

As previously stated, the output of a sensor array is a matrix comprising multivariate data, which is required to use the chemometric techniques to extract the relevant information that allows for the performance of (un)supervised qualitative (e.g., classification and discrimination) or quantitative studies. In general, the Enose is used for qualitative analysis, and the most commonly used multivariate techniques are principal component analysis (PCA), partial least square (PLS), self-organizing maps (SOM), hierarchical cluster analysis (HCA), artificial neural networks (ANNs), linear discriminant analysis (LDA), and support vector machines (SVMs) [18,21]. Few quantitative studies have been reported, and the multivariate techniques applied were multiple linear regression (MLR), the extreme learning machine model (ELM) [15], a single-hidden-layer feedforward neural network, PLS [16,22], SVM, decision tree (DT), and ensemble learning algorithms such as the random forest (RF) and AdaBoost models [23].

In this study, a lab-made Enose was designed and built comprising a vacuum sampling system and a closed sensor array chamber. The sensor array had nine Figaro MOS gas sensors, which had already been applied in qualitative studies (e.g., extra virgin olive oils commercial classification [19]), which were used to quantify the amounts of organic compounds in mixed aqueous solutions. For this purpose, two sets of experiments were performed following an experimental design to obtain aqueous solutions with orthogonal concentrations of three alcohols (i.e., methanol, ethanol, and propanol) or of three chemical compounds with different functional groups (i.e., acetaldehyde, ethanol, and ethyl acetate). The compounds were selected considering their availability in the laboratory.

## 2. Materials and Methods

### 2.1. Calibrations with Individual Standard Solutions

To establish the dynamic concentration ranges to be studied for the 5 different organic compounds (i.e., methanol, ethanol, propanol, aldehyde, and ethyl acetate), independent calibrations were carried out using 11 or 12 aqueous solutions of increasing concentration (all compounds were miscible in water). In all prepared solutions, concentrations were controlled by measuring the masses of the volumes used. The concentrations of the calibration standard solutions of each compound varied between 0.14 and 311.8 g/L for methanol; 0.14 and 307.5 g/L for ethanol; 0.14 and 306.9 g/L for propanol; 0.018 and 56.4 g/L for acetaldehyde; 0.13 and 340.8 g/L for ethyl acetate.

### 2.2. Calibrations with Standard Mixed Solutions

In order to check the system's capability for simultaneously quantifying the amounts of 3 alcohols (i.e., methanol, ethanol, and propanol), mixed aqueous solutions were prepared following an experimental design (Table 1). The design resulted into 25 orthogonal aqueous solutions of the combined chemical reagents at 5 different concentrations for each compound studied (concentrations confirmed by measuring the masses of the volumes used). In the alcohol's mixture, the concentrations were codified in 5 levels, from −2 to 2, from the lowest to the highest, varying from 0.36 to 156.9 g/L, 0.1 to 60.3 g/L, and 1.2 to 55.8 g/L, for methanol, ethanol, and propanol, respectively.

**Table 1.** A fractional factorial design for the mixing of 3 compounds with 5 different concentrations to assure orthogonal concentrations.

| Assay No. | Coded Levels | | |
|:---:|:---:|:---:|:---:|
| | Compound 1 | Compound 2 | Compound 3 |
| 1 | 0 | 0 | 0 |
| 2 | 0 | −2 | −1 |
| 3 | −2 | −1 | −2 |
| 4 | −1 | −2 | 2 |
| 5 | −2 | 2 | 2 |
| 6 | 2 | 2 | 0 |
| 7 | 2 | 0 | −1 |
| 8 | 0 | −1 | 2 |
| 9 | −1 | 2 | −1 |
| 10 | 2 | −1 | 1 |
| 11 | −1 | 1 | 1 |
| 12 | 1 | 1 | 0 |
| 13 | 1 | 0 | 2 |
| 14 | 0 | 2 | 1 |
| 15 | 2 | 1 | 2 |
| 16 | 1 | 2 | −2 |
| 17 | 2 | −2 | −2 |
| 18 | −2 | −2 | 0 |
| 19 | −2 | 0 | 1 |
| 20 | 0 | 1 | −2 |
| 21 | 1 | −2 | 1 |
| 22 | −2 | 1 | −1 |
| 23 | 1 | −1 | −1 |
| 24 | −1 | −1 | 0 |
| 25 | −1 | 0 | −2 |

Mix of alcohols: compound 1—methanol; compound 2—ethanol; compound 3—propanol. Mix of functional groups: compound 1—acetaldehyde; compound 2—ethanol; compound 3—ethyl acetate.

A second study was performed using the 3 compounds with different functional groups (i.e., acetaldehyde, ethanol, and ethyl acetate) mixed in aqueous solutions, prepared with a similar experimental design (as described in Table 1). The concentrations varied between 0.006 and 0.624 g/L, 0.20 and 59.3 g/L, and 0.04 and 36.9 g/L for acetaldehyde, ethanol, and ethyl acetate, respectively. In this study, the concentration levels of acetaldehyde were low in order to verify if the system could detect its presence when mixed with ethanol and ethyl acetate in higher amounts.

### 2.3. Lab-Made Enose

The lab-made Enose device integrated a vacuum sampling, prepared with the purpose to perform a quantitative analysis (Figure 1). Nine MOS Figaro sensors were used in the sensitive system (a sensor array based on resistive sensors), and their specifications are listed in the Table 2 (each sensor specification is available at https://www.figarosensor.com/product/sensor/, accessed on 4 April 2022).

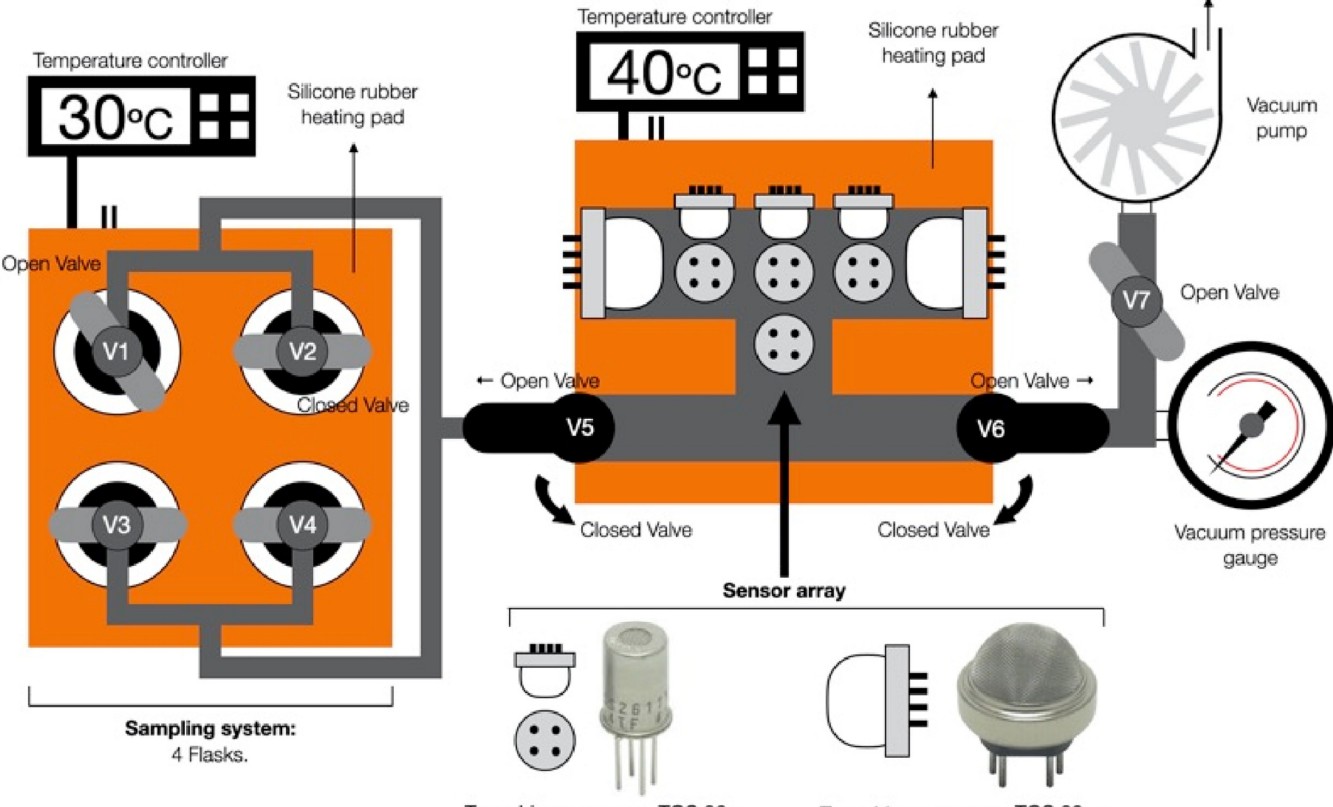

**Figure 1.** Schematic of the Enose built (in cleaning mode, when in valve V1, the flask was removed).

**Table 2.** List of MOS Figaro gas sensors used in the Enose sensor array device; $SnO_2$ was used as the basis of the sensing material.

| Code | Sensor | Tested Gases |
|------|--------|--------------|
| S1 | TGS 2600 B00 | Methane, CO, isobutane, ethanol, and $H_2$ |
| S2 | TGS 2602 | $H_2$, $NH_3$, ethanol, $H_2S$, and toluene |
| S3 | TGS 2610 C00 | Ethanol, isobutane, and $H_2$ |
| S4 | TGS 2611 C00 | Ethanol, $H_2$, isobutane, and methane |
| S5 | TGS 2610 D00 | Ethanol, $H_2$, and isobutane |
| S6 | TGS 2611 E00 | Ethanol, isobutane, $H_2$, and methane |
| S7 | TGS 2612 | Ethanol, methane, isobutane, and propane |
| S8 | TGS 826 A00 | Isobutane, $H_2$, ammonia, and ethanol |
| S9 | TGS 823 C12N | Methane, CO, isobutane, n-hexane, benzene, ethanol, and acetone |

The sensors were placed in a hermetic chamber that was controlled by manual gas valves. The system included a heated zone (30 °C) capable of accommodating 4 sample flasks and a multisensor system heating zone (40 °C), thermostatized using silicone rubber heating pads with temperature controllers. Each sample flask had a gas valve (plastic valve gas; valves V1 to V4 in Figure 1) as well as an entrance and exit to the multisensor system (mini brass ball valve gas; valves V5 to V6 in Figure 1). At the end of the Enose system, a vacuum pump (12 V DC diaphragm vacuum mini pump) was placed allowing either the cleaning of the system with the air flow pulled by the pump (when one of the sampling valves was open, as represented in Figure 1) or the preparation of the vacuum of the system when the sampling valves were all closed). The vacuum in the system was measured by a mini Dial Air Vacuum Pressure Gauge Meter Digital Manometer. Another plastic valve gas (valve V7 in Figure 1) was inserted between the vacuum gauge and the pump, which was used to confirm the absence of leaks in the vacuum. The connections among the Enose components (i.e., sample flasks, sensor system, vacuum gauge meter, and

pump) were made with transparent PVC tubes. All the electronic components, including the sensors, were connected to a Mean Well Swithing power supply (6 A, 12 V, 3 A). The sensors were also connected to a multiplexer Agilent Data Acquisition/Switch Unit model 34970A, which was controlled by the Agilent BenchLink Data Logger software, installed in a PC computer, for the sensor signals acquisition.

### 2.4. Pre-Procedures before Enose Analysis

The stabilization phase consisted of the sample pretreatment immediately prior to the analysis, where the samples were put under predefined conditions of time and temperature, aiming to achieve a liquid–vapor equilibrium (i.e., gas headspace and the aqueous solution). For this process, 2 mL of each aqueous sample were put in glass flasks of 25 mL and allowed to equilibrate for approximately 13 min at 30 °C before the analysis of the vapor phase. The volume of 2 mL was set considering the gas diffusion conditions defined by Henry's law. The time and temperature used for the stabilization period were set based on preliminary individual assays.

It was also necessary to ensure that the system was stable before performing any analysis. As the MOS sensors chosen required approximately 24 h to provide reliable stability for measurement, the equipment was always connected to the electrical current. This amount of time was necessary to activate the sensors' sensitive material, which needed temperatures in the range of 350–450 °C. Thus, before the first assay, this stabilization period must be implemented, although it is not required for subsequent assays, unless the system is disconnected from the electricity. Apart from the sensors' stabilization, each day, a time period of 30 min was required in order to ensure achievement of the set temperatures of the sampling and sensing zones (30 and 40 °C, respectively).

Before each new sampling, a constant signal baseline was required for each sensor to ensure the system's cleanliness, avoiding contamination by the previous sample. Therefore, external air was continuously injected for approximately 13 min (the same 13 min needed to ensure the liquid–vapor equilibrium), which allowed for obtaining stable signals, an important preliminary step affecting the vacuum formation and guaranteeing reproducible analysis conditions. Air was pumped at 25 mL/s, which was supplied to the system by opening one of the samples' valves.

### 2.5. Sample Analysis by Enose

The production of a vacuum environment of 0.35 bar, as the result of the closure of the sampling compartment valve together with the pump operation, provides a decrease in the sensors' resistance signals, helping in the exclusion of interferences that could be in external air flow. Beyond the positive influence in the signals, the vacuum also provides an increase in the cleaning of the sensitive material of the sensors. Thus, sample injection was given when valve V6 was closed (Figure 1), keeping the vacuum inside the analytical system, and the individual sample flask valve was opened, causing the suction of the vial headspace amount of gas into the sensor array's chamber. After closing the inlet (valve V5, Figure 1), the samples were analyzed in a closed space with a temperature different to that of the sample's zone. The sensors needed time for their signals to stabilize, which was fixed at 2.5 min, with signals (scans) being recorded every 4 s and the last scan used as the most representative. A temperature of 40 °C for the sensor array zone was chosen in order to assure that the suctioned content was mostly volatilized. The Agilent data logger read the resistance right on the sensors' surface. As referred above, the last signal point of each sensor was considered as the most stable to be used to create a "fingerprint" representing the volatile characteristics of the sample. Repetitions of the analyses were performed, having as an agreement criterion that the signal variation be less than 5%.

### 2.6. Statistical Analysis

All data treatments were performed using the statistic program R, version 3.2.0 (The R Foundation for Statistical Computing, Vienna, Austria), a free software environment for

statistical computing and graphics. The R statistical packages caret [24] and ggplot2 [25] were used.

A fractional factorial experimental design was applied to assure orthogonal concentrations of the 3 compounds in an aqueous solution. Pearson's correlation matrix was used to verify the degree of correlation between the signals from the Enose sensors.

A ridge regression model was applied to establish a fitting model between the concentrations and the sensors signals. This model was used when the data showed multicollinearity [26]. It is an extension of a linear regression, where the loss function is modified to minimize the complexity of the model by adding a penalty parameter that is equivalent to the square of the magnitude of the coefficients trying to minimize them, having the effect of shrinking the coefficients for those input variables with a low predictive performance. The data were split into a train data set (data from 18 aqueous solutions) and a test data set (data from 7 aqueous solutions). The predictor variables were centered, scaled, and exponentially transformed. The model's performance was evaluated using an internal validation variant, i.e., the cross-validation (CV) procedure with 6 folds and 10 repetitions for training, which implied the evaluation of the predictive performance of 100 different models. The average of the root mean square error (RMSE) and the mean absolute error (MAE) were used as predictive evaluation criteria. To interpret the sensors' coefficients within the model, a variable importance magnitude ranking was established. To visualize and evaluate the Enose's capability to quantify each compound's concentration in the aqueous solution, a simple linear regression model was established between the predicted model and real values for the training and testing data groups. The results were considered satisfactory if the linear regression parameters were close to the theoretical values [27–29]: "zero" (0) for relative standard error (RSE) and intercept; "one" (1) for slope and the determination coefficient. In addition, the confidence interval at 95% of the slope and intercept were used to statistically infer if they could be equal to the theoretic values of "one" and "zero", respectively.

## 3. Results

### 3.1. Calibrations with Individual Standard Solutions

The concentration intervals of the organic aqueous solutions used in the experimental designs were defined based on the Enose calibrations for each single compound (Henry's law application). Table 3 shows the concentrations and the sensor's signal range for each organic compound. As can be seen, ethyl acetate had the lowest mass dynamic interval followed by the acetaldehyde.

**Table 3.** Concentration ranges in the individual standard calibration and overall sensor resistance signal interval.

| Compound | Boiling Point, °C | $C_{min}$, g/L | $C_{max}$, g/L | Resistance Interval, Ohm |
|---|---|---|---|---|
| Methanol | 64.7 | 0.10 | 315.2 | [186; 33,374] |
| Ethanol | 78.4 | 0.14 | 307.5 | [194; 33,170] |
| Propanol | 97.0 | 0.14 | 306.9 | [225; 33,252] |
| Acetaldehyde | 20.2 | 0.018 | 56.4 | [106; 28,072] |
| Ethyl acetate | 77.1 | 0.024 | 34.8 | [75; 29,022] |

Figure 2 presents the response curves obtained for the concentration intervals studied (in logarithmic scale). This figure shows a similar analytical trend of the sensors' response for all organic compounds and different dynamic ranges among the sensors. This is an advantage when working with a sensor array, as it is possible to gather sensors that have a good response for low, medium, and high concentration levels and, overall, define a more robust multivariate calibration curve. The three alcohols presented higher signals than acetaldehyde and ethyl acetate, which showed similar responses to each other. However, for the selected dynamic range, acetaldehyde had signals at low mass concentrations (the lowest boiling point), while for the other compounds, the signals already appeared

to be constant. Moreover, the variability found in the acetaldehyde's curves can be explained by the difficulty of preparing the solutions due to the fact of its low boiling point. Globally, all sensors showed satisfactory responses for the highest concentrations of all organic compounds.

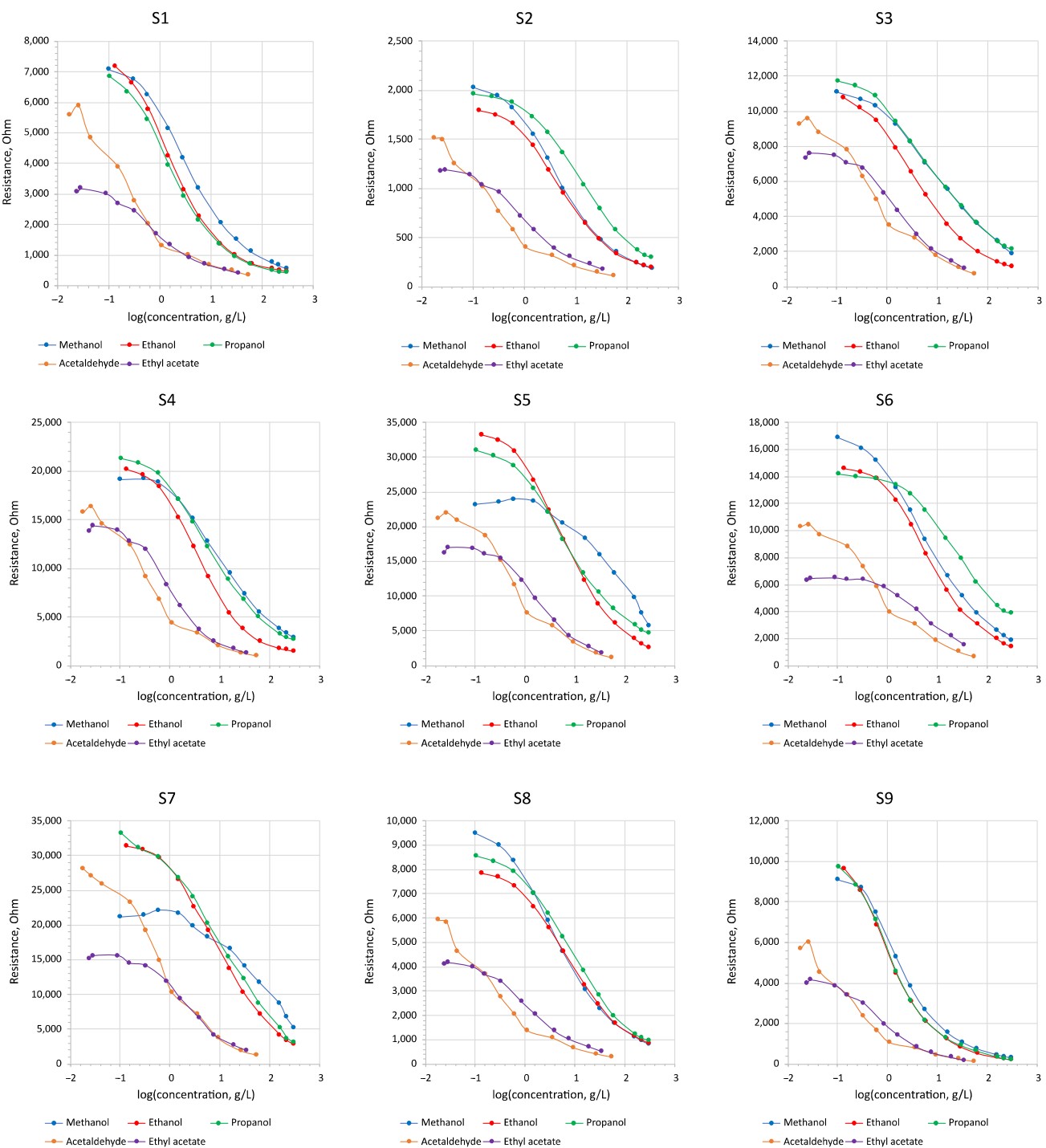

**Figure 2.** Sensors' response curves using aqueous solution of an individual organic compound (logarithmic scale of concentrations, g/L). S1-S9 are MOS Figaro gas sensors (description in Table 2) used in the Enose sensor array.

At low concentrations, it was found that some of the sensors appear to be in the non-detection zone, but others continued to be able to measure at these levels. This situation is

most evident in the measurement of ethyl acetate, where, for example, sensors S5, S6, and S7 showed approximately constant signals for concentration logarithms less than −0.05.

Usually, the sensors' signal variation in the calibration curves was S7 > S5 > S4 > S6 > S3 > S9 ≅ S8 ≅ S1 > S2. As can be seen in Figure 2, the sensors S7 and S5 were those that gave the widest signals, having obtained ranges around 33,000 Ohm in the independent calibrations.

Considering the variation in the sensors' signals in this study, it was decided to test the concentration intervals in a multivariate calibration study that would also allow the evaluation of the Enose's capability to measure low concentrations.

### 3.2. Calibrations with Standard Mixed Solutions

The experimental design for establishing the models to quantify the contents of the three alcohols in the mixed aqueous solutions or the contents of the three organic compounds with different functional groups is described in Section 2.2. In the experimental design, the total mass content of the organic compounds was also considered. For the aqueous solutions with the three alcohols, the concentrations varied between 0.30 and 60 g/L of methanol; 0.15 and 60 g/L of ethanol; 1.0 and 60 g/L of propanol; 18.2 and 256.9 g/L for the total mass. For the aqueous solutions containing the three organic compounds with three different functional groups, the concentrations levels varied in the range of 0.005 to 0.65 g/L of acetaldehyde; 0.15 to 60 g/L of ethanol; 0.03 to 40 g/L of ethyl acetate; 0.66 to 94.1 g/L for the total mass. The low concentrations of acetaldehyde (which shows the lowest boiling point) were chosen to verify if it would be possible to detect it at these levels, when the remaining compounds were present at higher levels and, therefore, a masking effect could occur.

Overall, the Enose analysis of these solutions showed resistance signals from 200 to 7106 Ohm and 130 to 9436 Ohm for the two respective experimental designs. The sensors' signals showed a high degree of correlation, usually higher than 0.85, with the exception of sensors S5 and S7, which showed correlations lower than 0.79 with all the other sensors.

As referred previously, the experimental data were divided into training (18 samples) and testing (7 samples) data sets. The results from the model fitting for training, using the ridge regression, are listed in Table 4. Each model's performance was evaluated for a CV variant with 6 folds and 10 repetitions using the training data set, which resulted in an average predictive performance for 60 different models. The models showed satisfactory predictive performances, with the exception of acetaldehyde, as can be seen from the wide range of values of the coefficient of determination obtained in the cross-validation ($R^2$ of $0.54 \pm 0.41$). The penalty lambda value of 0.0028 produced the lowest test mean squared error (MSE) for the cross-validation with 6 folds and 10 repetitions, which allowed for the selection of the best model.

**Table 4.** Ridge regression performance parameters for the cross-validation with 6 folds and 10 repetitions.

| Compound | Lambda | RMSE | MAE | $R^2$ |
|---|---|---|---|---|
| **Mixture of 3 alcohols** | | | | |
| Methanol | 0.0028 | $17.37 \pm 15.33$ | $14.08 \pm 11.29$ | $0.98 \pm 0.05$ |
| Ethanol | 0.0028 | $10.49 \pm 5.04$ | $8.93 \pm 3.94$ | $0.84 \pm 0.26$ |
| Propanol | 0.0028 | $10.72 \pm 7.49$ | $8.93 \pm 5.64$ | $0.88 \pm 0.18$ |
| Total | 0.0028 | $15.37 \pm 15.06$ | $12.33 \pm 11.42$ | $0.97 \pm 0.06$ |
| **Mixture of 3 organic compounds with different functional groups** | | | | |
| Acetaldehyde | 1.99 | $0.19 \pm 0.07$ | $0.17 \pm 0.06$ | $0.54 \pm 0.41$ |
| Ethanol | 0.0028 | $9.43 \pm 4.33$ | $8.36 \pm 3.98$ | $0.90 \pm 0.18$ |
| Ethyl acetate | 0.0028 | $2.88 \pm 1.38$ | $2.59 \pm 1.22$ | $0.96 \pm 0.08$ |
| Total | 0.0028 | $7.98 \pm 3.57$ | $6.99 \pm 3.33$ | $0.93 \pm 0.12$ |

RMSE—root mean square error; $R^2$—determination coefficient; MAE—mean absolute error.

Overall, the RMSE and MAE values were acceptable and of the same magnitude, while the mean determination coefficient, with two significant figures, was equal or higher than

0.84. To verify if the best model obtained allowed for the quantification of each organic compound's concentration in aqueous solution, parameters of the linear regression model established between the predicted and real values for the training and testing data sets are shown in Table 5. As can be seen, only acetaldehyde had poor predictive results for both training and testing data, as expected, due to the fact of its low concentrations. Overall, the results were satisfactory (Figures 3 and 4) with determination coefficients greater than 0.97 (close to one, the theoretical value), and the slope varied between 0.84 and 1.00 (close to one, the theoretical value). The intercept was not significant for some of the models, meaning that it can be considered equal to theoretical value of zero.

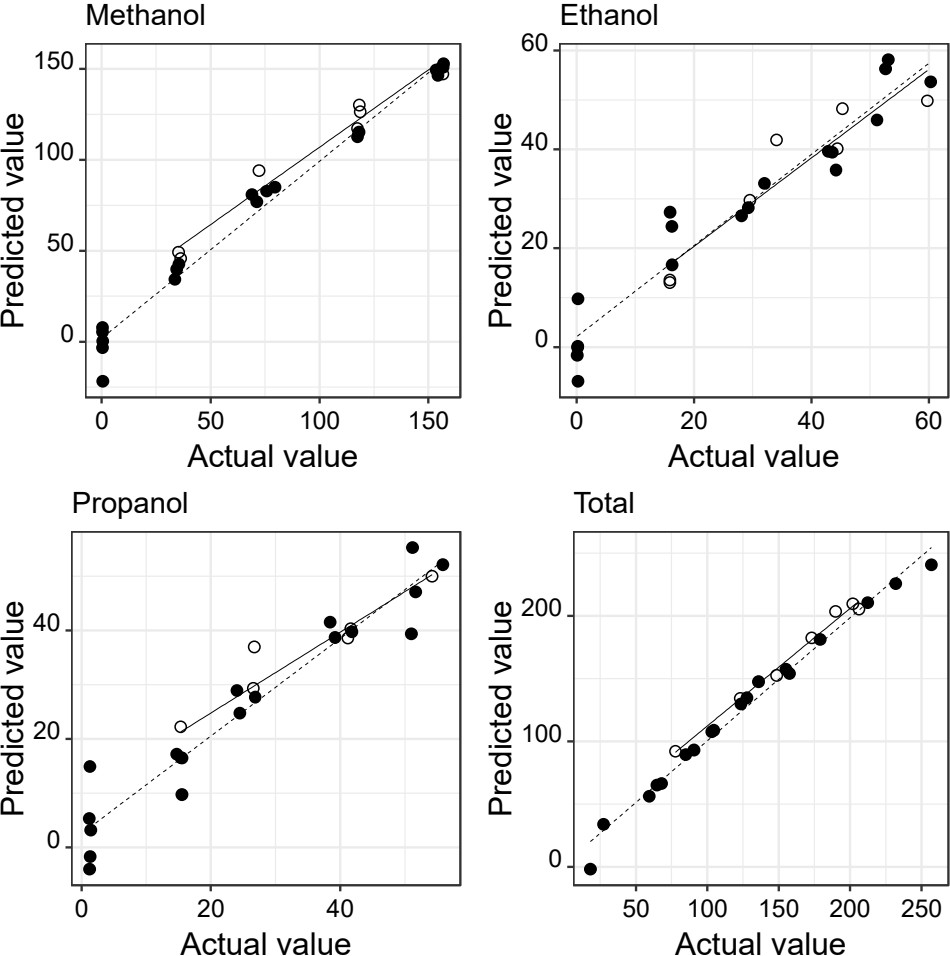

**Figure 3.** Enose predictive performance in relation to the expected concentrations (g/L) of the compounds in the mixtures of the 3 alcohol aqueous solutions and the total mass concentration (g/L) of these organic compounds; filled circles—training data; open circles—testing data.

**Table 5.** Parameters of the linear regression model established between the predicted model and real values for training and testing data sets.

| | Training Data Set | | | | | Testing Data Set | | | | |
|---|---|---|---|---|---|---|---|---|---|---|
| Compound | RSE | $R^2$ | *p*-Value | Slope (*p*-Value) | Intercept (*p*-Value) | RSE | $R^2$ | *p*-Value | Slope (*p*-Value) | Intercept (*p*-Value) |
| Mixture of 3 alcohols | | | | | | | | | | |
| Methanol | 7.90 | 0.993 | <0.001 | 0.99 ± 0.02 (<0.001) | Ns (0.528) | 8.05 | 0.966 | <0.001 | 0.85 ± 0.07 (<0.001) | 21.88 ± 7.29 (0.030) |
| Ethanol | 5.61 | 0.974 | <0.001 | 0.97 ± 0.04 (<0.001) | Ns (0.342) | 5.47 | 0.981 | <0.001 | 0.95 ± 0.05 (<0.001) | Ns (0.674) |
| Propanol | 5.33 | 0.972 | <0.001 | 0.96 ± 0.04 (<0.001) | Ns (0.231) | 5.58 | 0.978 | <0.001 | 1.01 ± 0.06 (<0.001) | Ns (0.052) |
| Total | 7.91 | 0.997 | <0.001 | 1.00 ± 0.01 (<0.001) | Ns (0.587) | 6.91 | 0.999 | <0.001 | 1.04 ± 0.02 (<0.001) | Ns (0.050) |
| Mixture of 3 organic compounds with different functional groups | | | | | | | | | | |
| Acetaldehyde | 0.06 | 0.07 | 0.303 | 0.10 ± 0.09 (0.303) | 0.16 ± 0.02 (<0.001) | 0.04 | 0.02 | 0.786 | −0.03 ± 0.09 (0.786) | 0.23 ± 0.03 (<0.001) |
| Ethanol | 4.05 | 0.957 | <0.001 | 0.84 ± 0.04 (<0.001) | 4.42 ± 1.54 (0.011) | 2.01 | 0.973 | <0.001 | 0.72 ± 0.05 (<0.001) | 5.41 ± 1.90 (0.036) |
| Ethyl acetate | 1.56 | 0.995 | <0.001 | 0.99 ± 0.02 (<0.001) | Ns (0.591) | 1.14 | 0.998 | <0.001 | 1.07 ± 0.02 (<0.001) | Ns (0.411) |
| Total | 4.36 | 0.966 | <0.001 | 0.89 ± 0.04 (<0.001) | 4.72 ± 2.10 (0.039) | 2.26 | 0.986 | <0.001 | 0.83 ± 0.04 (<0.001) | 6.57 ± 2.52 (0.048) |

RSE—relative standard error; Ns—not significant at the level of 0.05; *p*—*p*-value at the level of 0.05.

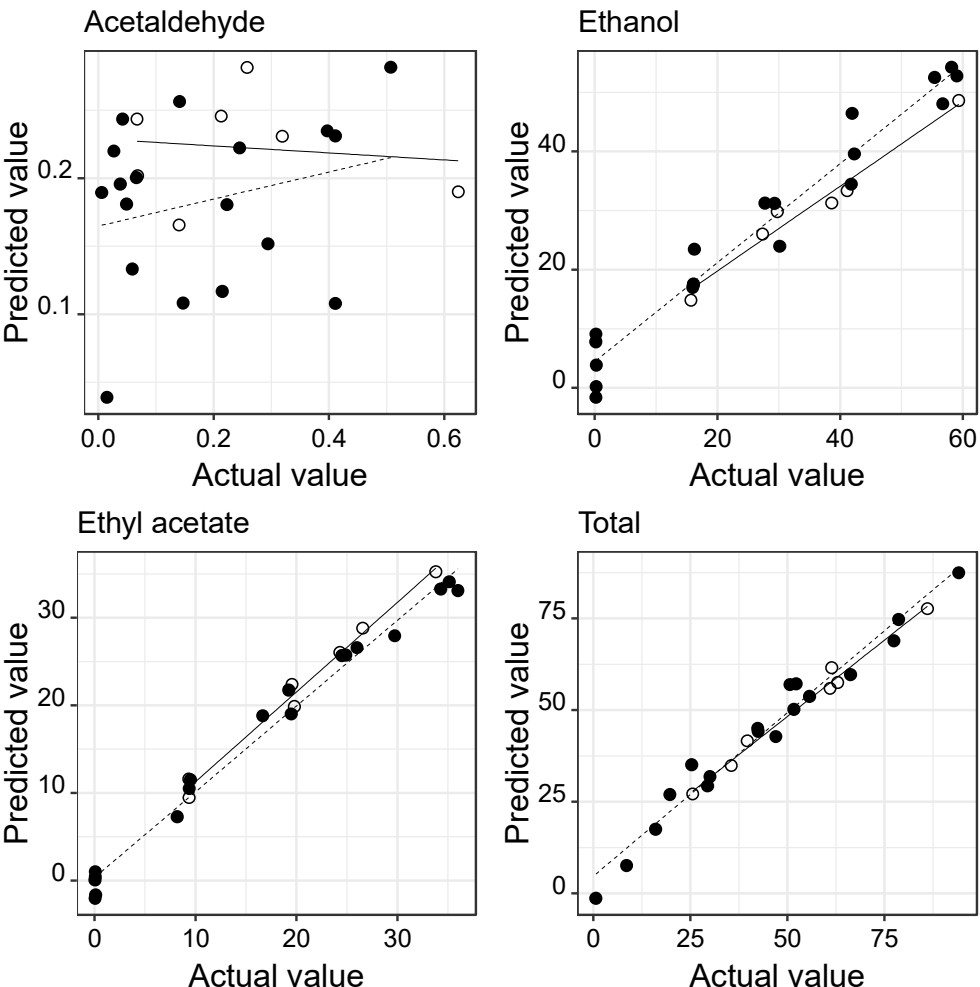

**Figure 4.** Enose predictive performance in relation to the expected concentrations (g/L) of the compounds in the mixtures of the 3 functional groups and the total mass concentration (g/L) of these organic compounds; filled circles—training data; open circles—testing data.

Moreover, similar results were obtained for the test data set, confirming the robustness of the models obtained for predicting the concentrations of the organic compounds in the mixed aqueous solutions, which were not used to establish the respective model. Since ethanol was analyzed in the two experimental designs, ethanol was present in both experimental designs (mixture of alcohols and mixture of organic compounds with different functional groups), which allowed to verify that the sensor array had a coherent predictive response for this organic compound, even when the solution matrix was different. The best results were obtained for the total mass of the organic compounds, inferring that the sensors measured mainly the global contents of the gases, since the sensor array was sensitive to all organic compounds studied (Figures 3 and 4).

The most important sensors in the prediction models (standardized coefficients) varied according to the organic compound analyzed. In general, all models included the signals recorded by the nine sensors. In the study of the mixture of the three alcohols, the following decreasing orders of importance were obtained in the prediction models of: methanol, S8 > S2 > S6 > S9 > S1 > S3 > S4 > S7 > S5; ethanol, S5 > S4 > S3 > S2 > S8 > S7 > S9 > S6 > S1; propanol, S1 > S2 > S9 > S8 > S7 > S5 > S6 > S4 > S3; total, S9 > S8 > S1 > S6 > S2 > S3 > S4 > S5 > S7. For the three organic compounds with different functional groups, the decreasing orders of the sensors' importance in the significant prediction models were: ethanol, S2 > S1 > S4 > S3 > S6 > S8 > S9 > S5 > S7; ethyl acetate, S9 > S4 > S3 > S8 > S2 > S1 > S5 > S6 > S7; total, S1 > S2 > S4 > S3 > S6 > S5 > S8 > S9 > S7. By comparing the ethanol

prediction models (also, the total) obtained in the two studies, it was found that the sensors had differing importance in the two models, indicating that the models were different and highlighting the influence of the matrix under analysis.

Regarding the low levels of acetaldehyde concentration, no quantitative estimation or prediction model could be established. Considering the objective of verifying whether it was possible to detect the presence of acetaldehyde in the samples, the analytical data showed to be random at the concentration levels tested.

Overall, the results show that the Enose must be calibrated and tested for each specific situation where it is intended to be applied, since the quantitative proportion in a given matrix influences the final results. Furthermore, Figures 3 and 4 show that low levels were a problem in the predictive models, with negative values obtained for the lowest concentration levels. Considering that acceptable results were achieved using a wide range of concentration levels, including low concentration levels, to demonstrate the analytical robustness of the Enose constructed, it can be inferred that this is a useful analytical tool for studying the contents of volatiles in different matrices. The architecture of the built system showed clear advantages in the analysis of aqueous solutions with volatile compounds, the results of which can be compared with the quantification works presented by other authors, which are scarce in number.

A similar work was presented by Ma et al. [22] that used an Enose (including four commercial MOS gas sensors, a temperature sensor, and a humidity sensor) to quantify six individual toxic gases (multiple indoor air contaminants: hydrogen sulfide, carbon monoxide, ammonia, toluene, formaldehyde, and acetone) and three kinds of binary gas mixtures by PLS technique. The average values of $R^2$ for the training and test samples were equal to 0.957 and 0.927, respectively. For the ternary gas mixtures of the present work, using a linear regression technique (ridge regression, less complex than PLS), the mean values of $R^2$ for the training and test data groups were 0.979 and 0.983 (not considering the acetaldehyde results), respectively. In addition, Wu et al. [11] used an Enose with a 14 metal oxide sensor array for quantitative monitoring of five highly flammable liquids (i.e., ethanol, tetrahydrofuran, turpentine, lacquer thinner, and gasoline) by selecting the most adequate sensor. The predictive model was obtained by polynomial or power regression analysis and the average errors were 9.1–18.4%. In the present work, the relative standard error obtained was lower than 8.1% for the training and test data sets. In the analysis of ethanol in real samples, Jordan Voss et al. [3] used 13 gas sensors (metal oxide semiconductor) that were applied to a training group of standard solutions prepared with distilled water and ethanol with 15 predetermined alcohol contents. A multiple linear regression model was obtained and applied in the detection of alcohol content in seven different commercial beverages, i.e., beers (test group). The results presented a determination coefficient of 0.888 and an RMSE of 0.45 in the test group, while an extreme learning machine model (ELM), with single-hidden-layer feedforward neural network, presented an RMSE of 0.33. Other works have shown that Enoses may be able to quantify several compounds present in complex mixtures, such as food products, although they were works where only estimation models were obtained. For example, Cui et al. [16] presented a work on the discrimination of American ginseng and Asian ginseng using an Enose (using seven MOS sensors) and gas chromatography-mass spectrometry. Some selected constituents (i.e., α-farnesene, β-panasinsene, α-gurjunene, alloaromadendrene, γ-muurolene, γ-selinene, viridiflorol, α-copaene, and octanal) showed estimation fitting models with $R^2$ values between 0.83 and 0.94. From the analysis of real samples with Enoses, it is expected to obtain a profile of signals with the influence of the matrix, increasing the complexity of the multivariate treatment. A work by Wijaya et al. [23] provides an example of how more complex regression models can be applied to fit the analytical data (estimation models) such as the SVM and an ensemble learning approach using DT, RF, and AdaBoost for Enose signal processing. Beef quality assessment was estimated with several multivariate techniques presenting $R^2$ values higher than 0.99 and an MSE lower than 0.04. However, in the present work, the complexity of the experimental design was reduced compared to

using real samples and, therefore, the simplest possible predictive model should be used to ensure simplicity in data handling and robustness of predictions.

These quantitative works make it possible to verify that the Enoses used differed in their architecture and in the sampling and analysis procedures. The Enose used by Ma et al. [22] comprised a mixture chamber of gases that was connected to a sensor array chamber with a fan. The target gas mixture was imported into the sensor array chamber for 5 min for measurement, and afterwards, the sensor array was exposed to clean air for 10 min again to recover the baseline and wash the sensors. Wu et al. [11] used an Enose with a separate evaporation chamber full of fresh air into which a liquid sample was injected through the sampling hole by a pipette. After closing the chamber, a fan kept drawing the gases evaporated from the sample into the sensor array until the sample evaporated completely. Finally, a lid opened and the sensor array was exposed to fresh air until reaching the baseline in order to start a new measurement. The Enose applied by Jordan Voss et al. [3] presented a sensor array inserted in a box with holes in the cover and sides and a cooler to circulate air (positioned above the sensors, with constant flow that allowed to homogenize the air reaching the sensors while maintaining a stable temperature), which was pulled inside the device. A similar device was used by Wijaya et al. [23]: a box with two chambers, with the gas sensor array in the first and the control box with a wireless communication module in the second. Since this device was used to analyze meat, the gas sensor signal from the box was stored continuously for approximately 2220 min in each experimental round and the cleaning used a high-speed fan for a 3 to 6 h of interval between samples to remove any lingering odor residue caused by the previous experiment. In the work of Cui et al. [16], the commercial FOX 4000 Enose device (Alpha MOS, Toulouse, France) was used, having gas sensors located in three temperature-controlled chambers (high temperature and zero humidity) and using a purified air generator to provide carrier gas for cleaning sensors. This device allows to control the sample temperature, which in the work presented was 1 h of stabilization at 60 °C.

Compared with the architectures of these Enoses, the present Enose, made with vacuum sampling, had a design that helped in cleaning the sensors, used temperature control in the sampling and gas analysis, and allowed for the aspiration of the volatile sample to the chamber with the set of sensors in a controlled manner and without interference from the cleaning gas.

## 4. Conclusions

The built Enose lab made with vacuum sampling had a design that allowed for the simultaneous quantitative analysis of three compounds present in an aqueous solution. The volatiles sampling procedure (gas equilibrium in aqueous solution defined by Henry's law) showed to be adequate for the quantification study. Even so, the variability in the analytical data can also be explained by the fact that very volatile organic compounds were used. It can also be inferred that the Enose would allow for even better results if the low concentration levels were increased. Furthermore, the inclusion of new sensors with sensitivities to different organic compounds not covered by those used in the sensor array in this study may improve the response in analyses of more complex matrices.

**Author Contributions:** G.G.T., investigation; A.M.P., investigation and funding acquisition; P.G., investigation; L.E., conceptualization and writing; F.M.-P., formal analysis and methodology; C.G.-C., conceptualization and writing—original draft; M.L.R.-M., conceptualization and writing—review and editing; L.G.D., investigation, methodology, supervision, and writing—review and editing. All authors have read and agreed to the published version of the manuscript.

**Funding:** This research received no external funding.

**Institutional Review Board Statement:** Not applicable.

**Informed Consent Statement:** Not applicable.

**Data Availability Statement:** Not applicable.

**Acknowledgments:** The authors are grateful to the Foundation for Science and Technology (FCT, Portugal) and FED-ER under Programme PT2020 for financial support by national funds FCT/MCTES to CIMO (UID/AGR/00690/2019) and SusTEC (LA/P/0007/2020).

**Conflicts of Interest:** The authors declare no conflict of interest.

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
