# Peer review of "Enose Lab Made with Vacuum Sampling: Quantitative Applications"

_chemosensors, doi:10.3390/chemosensors10070261_

Round 1

Reviewer 1 Report

The paper is presenting an interesting topic on (Enose) developed with vacuum sampling and a sensor array together with regression models. R2 values greater than 0.95 and the sensors measured mainly the global contents of the vapor phases. the few comments and quetions below are essential before we accept this paper:

1) the equipment was always connected to the electrical current during stabiltiy analysis of the system. is this not creating artefacts in the data? this might be important when you raise the temperature to over 400C.

2) variability found in the acetaldehyde’s curves due to difficulty in preparinf the solutions. how much this affects on the credibility of the sensor response. have you considered the reproducibility of the same?

3)exception of acetaldehyde in the results of Fig. 1 and 2 requires to be explained in correlation to results presented in [11].

4) citation is essential to https://doi.org/10.3390/coatings12060823

when authors take my comments in i can reconsider my decision.

Author Response

We thank the reviewer for his attention, appreciation and contribution to the improvement of the submitted article.

The paper is presenting an interesting topic on (Enose) developed with vacuum sampling and a sensor array together with regression models. R2 values greater than 0.95 and the sensors measured mainly the global contents of the vapor phases. the few comments and questions below are essential before we accept this paper:

1) the equipment was always connected to the electrical current during stabiltiy analysis of the system. is this not creating artefacts in the data? this might be important when you raise the temperature to over 400C.

When performing analyzes, the equipment was turned on the day before to ensure that the internal temperatures of the system were constant. This care was relevant mainly in the stabilization of the temperature of the gas sensors, a necessary step to have stable and reproducible signals. With this procedure, no extraneous signals were obtained in the obtained data matrices.

2) variability found in the acetaldehyde’s curves due to difficulty in preparinf the solutions. how much this affects on the credibility of the sensor response. have you considered the reproducibility of the same?

The reviewer is right in that the preparation of solutions with organic solvents was an important point of the work. To ensure greater accuracy in the preparation of solutions, these were controlled with measurements of the masses of the volumes of solvents measured for the volumetric flasks. Considering this procedure, the reproducibility of measurements was evaluated as acceptable (variation of analytical signals of solutions was less than 8%).

However, as mentioned by the reviewer, acetaldehyde showed greater variability mainly because it is a more volatile solvent and, therefore, with the possibility of losses both in the preparation of the solutions and in their measurement with the Enose analytical system (volume measurement with a micropipette and transfer to an Enose sampling vessel).

In light of the reviewer's comment, a new sentence and text has been included in the new manuscript stressing this point.

New phrase was introduce in page 2, line 83: “In all prepared solutions, concentrations were controlled by measuring the masses of the volumes used.”; New text was introduce in page 2, line 94: “… (concentrations confirmed by measuring the masses of the volumes used).”

3) exception of acetaldehyde in the results of Fig. 1 and 2 requires to be explained in correlation to results presented in [11].

Table 4 presents an average determination coefficient of 0.54+-0.41 which represents a correlation coefficient of 0.73. But, this result is associated with a high variation of the experimental points (graph not shown), failing completely as an estimation model and, therefore, also as a prediction model.

To clarify this issue, new text was introduced in the page 8 line 280: “ …, with the exception of acetaldehyde, as can be seen from the range of values of the coefficient of determination obtained in the cross-validation (R2 of 0.54± 0.41).”

4) citation is essential to https://doi.org/10.3390/coatings12060823

The article https://doi.org/10.3390/coatings12060823 is “Synthesis and Characterization of Highly Photocatalytic Active Ce and Cu Co-Doped Novel Spray Pyrolysis Developed MoO3 Films for Photocatalytic Degradation of Eosin-Y Dye”

I read the article indicated and I think that there must have been a mistake in the reference as I do not see any relationship with the present work. I ask the reviewer to explain what content I should insert to improve the article and to confirm the reference to be introduced.

Reviewer 2 Report

Dear Autors,

the article deals with electronic nose (Enose) with vacuum sampling and a sensor array, comprising 9 metal oxide semiconductors Figaro gas sensors. I have the following comments and questions:

- How the stabilization time and temperature were determined?

- How did the bank clean up? Was it uniform for all samples? Just by air? Is it enough?

- Are there ways to improve reliability at lower concentrations?

- How was the number of measurements determined?

- In the discussion, I miss the comparison of yours proposal with cited literary sources. Has your solution strengths and weaknesses compared to the cited sources. What does it add to the subject area compared with other published material? Highlight the benefits.

- Why Henry's law was used?

- What was the sensitivity, response, limits, accuracy of the sensors (sensor specifications)?

Thank you and have a nice day.

Author Response

We thank the reviewer for his attention, appreciation and contribution to the improvement of the submitted article.

Dear Autors,

the article deals with electronic nose (Enose) with vacuum sampling and a sensor array, comprising 9 metal oxide semiconductors Figaro gas sensors. I have the following comments and questions:

- How the stabilization time and temperature were determined?

The temperature of sample’s vessels was selected so as not to saturate the sensor signal and allow samples to stabilize in an adequate time. We consider that the equilibrium between the aqueous phase and the gas phase should be at a low temperature so that the gas phase has, mainly, the contribution of organic compounds and that water is little present. With trial and error, the temperatures used in the work were selected. The stabilization time is more than enough to stabilize the sample’s temperature and was chosen mainly to allow continuous analysis of samples without time’s pressure in the execution of the steps associated with the analysis procedure. In other words, the 13 minutes of stabilization allow the sample to reach phase equilibrium and clean the Enose internal system for a new analysis.

In page 5, line 149, the sentence presented is to give this indication, in which tests were carried out to define these parameters.

- How did the bank clean up? Was it uniform for all samples? Just by air? Is it enough?

The cleaning procedure involved the use of air which allowed to remove the organic gases from the Enose system and obtain stable signals (uniform procedure performed between all analyzes). However, the main step in preparing the Enose system for a new analysis was the suction of inner atmosphere present inside the sensor array chamber due to the formation of a new vacuum, which in turn would allow the introduction of the new sample present in one of the vessels of the sampling system.

New text was introduced in page 5 and line 163 to clarify this issue: … , an important preliminary step to effect the vacuum formation and guaranteed reproducible analysis conditions.”.

Also, a phrase in page 5, line 168 refers the importance of making the vacuum: “The production of a vacuum environment of 0.35 bar, as the result of the closure of the sampling compartment valve together with the pump operation, provides a decrease in the sensors’ resistance signals, helping in the exclusion of interferences that could be in external air flow.”

- Are there ways to improve reliability at lower concentrations?

The present work focused on confirming that the architecture of the Enose built, based on vacuum sampling, allowed quantitative analyzes of volatile compounds. This work is relevant considering that, in general, Enose equipments are used for qualitative applications. Therefore, inexpensive sensors with generic characteristics were selected.

So, these considerations were referred in the conclusion section of the manuscript. An alternative to improve the performance of Enose for low concentrations will be to select sensors with better performance; another, would be to increase the concentration of the compound or mixture of compounds in the aqueous solution, which would increase the levels of volatiles in the gas phase.

- How was the number of measurements determined?

The number of measurements of each solution was dependent on the agreement criterion of the final signal obtained, that is, if the average signal had a variability of less than 5% (percentage relative standard deviation). In general, this criterion was always fulfilled in the measurements carried out showing the reproducibility of the analyzes with Enose.

A new phrase was introduced regarding this issue (page 5, line 182): “Repetitions of the analyzes were performed having as an agreement criterion the signal variation being less than 5%.”

- In the discussion, I miss the comparison of yours proposal with cited literary sources. Has your solution strengths and weaknesses compared to the cited sources. What does it add to the subject area compared with other published material? Highlight the benefits.

We believe that the question raised is relevant and helps to improve the quality of the article. New text was introduced in the article discussion (page 9, line 384):

“These quantitative works make it possible to verify that the Enose used differ in their architecture, in the sampling and analysis procedures. The Enose used by Ma et al. [22] comprises a mixture chamber of gases that it is connected to a sensor array chamber with a fan. The target gas mixture is imported into the sensor array chamber for 5 min for measurement and afterwards, the sensor array is exposed to clean air for 10 min again to make it recover the baseline and wash the sensors. Wu et al. [11] used an Enose with a separate evaporation chamber full of fresh air, where a liquid sample was injected into through the sampling hole by a pipette. After closing the chamber, a fan keeps drawing the gases evaporated from the sample into the sensor array till the sample evaporates completely. Finally, a lid opens and the sensor array is exposed to fresh air until reaching the baseline, in order to start a new measurement. The Enose applied by Jordan Voss et al. [15] presents a sensor array inserted in a box with holes in the cover and sides and a cooler to circulate air (positioned above the sensors, with constant flow that allowed to homogenize the air reaching the sensors, while maintaining a stable temperature), which is pulled inside the device. Similar device was used by Wijaya et al. [23], a box of two chambers, having in the first the gas sensor array and, in the second chamber, the control box with wireless communication module. Since this device was used to analyse meat, the gas sensor signal from the box was stored continuously for about 2220 min in each experiment round and the cleaning used a high-speed fan and 3 to 6 h of interval between samples to remove any lingering odor residue caused by the previous experiment. In the work of Cui et al. [16], the commercial FOX 4000 Enose device (Alpha MOS, Toulouse, France) was used, having gas sensors located in three temperature-controlled chambers (high temperature and zero humidity) and using a purified air generator to provide carrier gas for cleaning sensors. This device allows to control de sample temperature, which in the work presented was 1 hour of stabilization at 60 ºC.

Comparing with the architectures of these Enose devices, the present Enose made with vacuum sampling has a design that helps in cleaning the sensors, uses temperature control in the sampling and gas analysis, allows the aspiration of the volatile sample to the chamber with the set of sensors in a controlled manner and without interference from the cleaning gas.”

- Why Henry's law was used?

Our research laboratory is not equipped for gas analysis and, therefore, we had to come up with strategies on how to establish the relationship between the amount of organic compounds in the headspace and in the solution, which is an aqueous solution defined by the Henry's law. In practical terms it justifies the quantification models. This procedure can also open the range of applications of Enose, either in other quantitative applications or in qualitative applications, such as in food analysis.

- What was the sensitivity, response, limits, accuracy of the sensors (sensor specifications)?

Regarding the sensor specifications, the manufacturer makes an exposition of the general characteristics of each sensor, being relevant information for the article. Although a summary was made in Table 2, due to the question posed, we consider it relevant to indicate to readers where the information is available.

Therefore, new text has been introduced on the page 4 and line 117: (each sensor specification is available in https://www.figarosensor.com/product/sensor/).